# Relationship between Fatigue and Self-Perception of Constipation in Community-Dwelling Older Adults during the COVID-19 Pandemic

**DOI:** 10.3390/ijerph19148406

**Published:** 2022-07-09

**Authors:** Chisato Hayashi

**Affiliations:** Research Institute of Nursing Care for People and Community, University of Hyogo, 13-71, Kitaoji-cho, Akashi 673-8588, Japan; chisato_hayashi@cnas.u-hyogo.ac.jp; Tel.: +81-78-925-9653

**Keywords:** self-perception of constipation, fatigue, COVID-19, community-dwelling older adults

## Abstract

Fatigue and constipation are common symptoms among community-dwelling older adults; however, no studies have explored the relationship between both symptoms in said group. Therefore, this study aimed to examine the relationship between fatigue and self-perception of constipation in community-dwelling older adults during the COVID-19 pandemic. A cross-sectional survey was conducted among 97 older people (response rate: 73.2%) between July and November 2021. Backward–forward stepwise linear regression was performed to identify possible predictors of fatigue among the explanatory variable (self-perception of constipation) and possible confounders, which included (a) age, (b) motor fitness scale, (c) economic satisfaction, (d) subjective memory impairment, (e) subjective health, (f) depression, (g) living alone or not, (h) sex, and (i) frequency of exercise. The intercept of the fatigue score was 42.48 points (95% CI = 32.40 to 49.99, *p* < 0.001). The results showed that the fatigue score in community-dwelling older adults who had self-perception of constipation was significantly lower (i.e., they had higher fatigue; B = −4.49, 95% CI = −6.58 to −2.40, *p* < 0.001) compared to those who did not have self-perception of constipation. Therefore, there is a need to develop self-management strategies that older people can use to improve the self-perception of constipation.

## 1. Introduction

Although there are many studies on fatigue in patients with specific diseases, such as cancer [1,2], chronic kidney disease [3], rheumatic disease [4], and Parkinson’s disease [5], few studies have focused on fatigue in community-dwelling older adults. Fatigue is one of the most commonly described symptoms in community-dwelling older adults, with a prevalence of approximately 47.9% [6].

Constipation is also a common problem in older people; however, it is often neglected or rarely discussed because it is considered a private matter [7]. Constipation’s primary symptoms are reduced defecation frequency and experiencing strain when defecating [8]. A recent systematic review estimated that the prevalence of constipation is 33.5% among adults aged >60 years [9]. In addition, a study reported that, during the COVID-19 pandemic, one-fourth of the population developed new-onset functional constipation symptoms and lower water consumption due to related lockdown measures and the resulting reduction in physical activity [10]. Moreover, constipation has been reported to have a significant impact on quality of life [11] and associated with depression [12].

A recent literature review on fatigue in community-dwelling older adults reported various factors related to fatigue, including biological (inflammation, energy, pain, adiposity, low hemoglobin, and tooth loss), psychological (depressive symptoms and stress), behavioral (sleep, physical activity, and malnutrition), and social factors (social support) [13]. A study of patients with lung cancer receiving platinum-based chemotherapy found that reduced food intake and fatigue were the main factors affecting functional constipation [11].

A common feature of fatigue and constipation is that they are experienced differently by different people, and there is a divergence from the clinical diagnosis. Fatigue, in particular, is difficult to observe objectively and is subjective. However, regarding constipation, it is not only possible for an individual to understand the problem by keeping a stool diary, but it is also easy to share the problem with a healthcare provider [14,15].

A previous study found another definition of constipation. They reported that 50% of patients defined constipation differently from its medical definitions [16]. Additionally, a recent study reported that the percentage of people in Japan who considered themselves constipated was 28.4%, and of these, 52.2% met the diagnostic criteria for Rome III. Furthermore, 52.2% of the self-diagnosed constipation respondents were diagnosed with chronic functional constipation, 23.8% with irritable bowel syndrome (IBS), and the remaining 24% with other diseases [17]. The “self-perception of constipation” is an easy question for healthcare workers to ask and for community members to answer. If self-perception of constipation is found to be related to fatigue, which is commonly observed in other diseases, it could be a candidate for active adoption as a primary healthcare assessment item.

To the best of my best knowledge, there have been no studies on the relationship between fatigue and self-perception of constipation in community-dwelling older adults. In Japan, the COVID-19 pandemic brought community group activities to a halt. However, there were periods when such activities could be implemented between the pandemic waves, such as in July, October, and November 2022. We conducted a cross-sectional survey. This study analyzed valuable data from elderly people living in the community who participated in community activities during those periods. Therefore, this study aimed to examine the relationship between fatigue and self-perception of constipation in community-dwelling older adults during the COVID-19 pandemic.

## 2. Materials and Methods

### 2.1. Participants

This study included 71 community-dwelling older adults who attended a health course at a local institution. The sample size was determined using R software’s “pwr” package [18]. A threshold alpha was set at α = 0.05; moreover, considering an effect size of 0.26, the minimum sample size was estimated as 60 participants.

### 2.2. Procedure

The data collection period ranged from July to November 2021. We distributed questionnaires to 97 older people and collected 71 questionnaires (response rate: 73.2%). Backward–forward stepwise linear regression was used to identify possible predictors of fatigue out of the following variables: the explanatory variable (self-perception of constipation) and possible confounders, which included (a) age, (b) motor fitness scale (MFS) score, (c) economic satisfaction, (d) subjective memory impairment (SMI), (e) subjective health, (f) depression, (g) living alone or not, (h) sex, and (i) frequency of exercise.

### 2.3. Data Analysis

All statistical analyses were conducted using R software, version 4.1.0 (Vienna, Austria) [19]. The “car” package was used to display the variance inflation factor and model selection [20]. At each step, variables were removed based on the Akaike information criterion, which is a method of removing variables sequentially from those with large Pr (>|t|) values; the operation of removing variables is repeated until the optimal model is selected. In the regression analysis, the five covariates were statistically significant. Statistical significance was set at *p* < 0.05.

### 2.4. Instruments

#### 2.4.1. Outcome Variable

Fatigue was assessed using the Functional Assessment of Chronic Illness Therapy—Fatigue scale (FACIT-F) (score range: 0–52), which was translated into Japanese by Yoshimura et al. [21]. The original version has high sensitivity (0.92) and reasonable specificity (0.69) [22]. The FACIT-F score was found to be stable (test–retest r = 0.87) and internally consistent (coefficient alpha range = 0.95) [23]. It contained nine questions inquiring about the extent of fatigue during the past seven days on a 5-point Likert-type scale (ranging from 0 (“not at all”) to 4 (“very much”)). Before the 13-item response data from the FACIT-F were analyzed, all negatively worded items were reversed, as instructed in the manual [24]. Therefore, higher scores represent less fatigue and lower scores represent more fatigue.

#### 2.4.2. Explanatory Variable

Self-perception of constipation was assessed by the question “Do you often have constipation?” [21]. Responses were recorded on a 3-point Likert-type scale (0 = no, 1 = sometimes, and 2 = yes). The participants who answered the above question with a ‘yes’ and ‘sometimes’ were considered to be ‘self-perception of constipation’ [25].

#### 2.4.3. Possible Confounders

As possible confounders, (a) age, (b) MFS, (c) economic satisfaction, (d) subjective memory impairment (SMI), (e) subjective health, (f) depression, (g) living alone or not, (h) sex, and (i) frequency of exercise were assessed.

The mean age of the entire group was 78.0 years old. Therefore, the intercept of the fatigue estimate showed a value of 78.0 years old. To evaluate participants’ physical abilities, we used the MFS, which comprises three subscales: mobility, muscle strength, and balance [26]. The MFS consists of 14 questions, wherein higher scores indicate better physical performance. Internal consistency was α = 0.92 and the test–retest reliability individual correlation was 0.92 [27].

Economic satisfaction was assessed using the following question: “How satisfied are you with your financial situation?” Responses were rated on a 5-point Likert-type scale (4 = very satisfied, 3 = sufficiently satisfied, 2 = neutral, 1 = somewhat satisfied, and 0 = not sufficiently satisfied). SMI was assessed using the following question: “Have your family or friends pointed out your memory loss? E.g., they have mentioned that you ask the same question repeatedly.” The responses were restricted to either “Yes” or “No.” This question was extracted from the Kihon Checklist, which was developed to identify individuals eligible for long-term care prevention projects in Japan [28].

Subjective health was one of the questions asked in the National Survey of the “Comprehensive Survey of Living Conditions.” Subjective health was assessed using the following question: “Currently, how is your health?” Responses were rated on a 5-point Likert-type scale (4 = very good, 3 = good, 2 = regular, 1 = poor, and 0 = very poor) [29]. Depression was assessed using the Geriatric Depression Scale-15 (GDS-15) [30]. The GDS-15 consists of 15 questions regarding how the respondent felt over the past week. Frequency of exercise was assessed using the following question: “How often do you usually exercise?” Responses were rated on a 4-point Likert-type scale (0 = never, 1 = hardly ever, 2 = occasionally, 3 = usually).

## 3. Results

### 3.1. Descriptive Statistics

Descriptive data are shown in Table 1. The study included 12 men and 59 women. Participants’ mean age was 78.3 ± 5.5 (mean ± SD) years old. The mean FACIT-F score was 40.2 ± 9.7, the mean MFS score was 10.6 ± 3.6, and the mean GDS score was 4.0 ± 3.9. Regarding the constipation question, 10 participants answered with a “yes” and 20 participants with “sometimes.” In this study, a total of 30 people who answered “yes” and “sometimes” were included in the constipation group. Twenty-one participants were living alone (29.6%) and nine participants reported subjective memory impairment (12.7%). Regarding the frequency of exercise, 37 participants usually exercised (52.1%), 26 participants occasionally exercised (36.6%), while 8 participants hardly ever exercised (11.3%). Concerning economic satisfaction, 17 participants were very satisfied (23.9%), and 28 participants were sufficiently satisfied (38.4%). Regarding subjective health, 4 participants were very good health wise (5.6%), while 13 participants were good (18.3%).

### 3.2. Regression Analysis

Table 2 presents the correlation analysis between fatigue and explanatory variables.

Table 3 presents the results of the single-regression analysis. Age, MFS score, depression, subjective health status, and self-perception of constipation were significantly associated with fatigue.

Next, multiple regression analysis was conducted (Table 4). Among the possible confounders, frequency of exercise (Stage 1), living alone or not (Stage 2), SMI (Stage 3), and subjective health sex (Stage 4) were removed from the best-fit model (Stage 5). Self-perception of constipation, age, MFS score, depression, and economic satisfaction remained in the final model.

The Durbin–Watson ratio was 1.385 in the final model (Stage 5). Plots of residuals vs. fitted, normal Q-Q, scale-location, and residuals vs. leverage were visually confirmed (see Appendix A). Residuals were normally distributed and Cook’s distance was less than 0.5.

The intercept of the fatigue score was 42.48 points (95% CI = 32.40 to 49.99, *p* < 0.001). The results showed that the fatigue score in community-dwelling older adults with self-perception of constipation was significantly lower (B = −4.49, 95% CI = −6.58 to −2.40, *p* < 0.001) compared to those who did not have self-perception of constipation (Table 4).

The fatigue score was significantly lower in older adults with depression (B = −1.39, 95% CI = −1.75 to −0.88, *p* < 0.001), those with high MFS scores (B = 0.99, 95% CI = 0.57 to 1.41), and those with high economic satisfaction (B = −1.75, 95% CI = −2.85 to −0.38, *p* < 0.01).

The variance inflation factors were all below 2.0, and there were no multicollinearity problems. The results of the ANOVA were significant, with a coefficient of determination (R^2^) of 0.78 and an adjusted R^2^ value of 0.76, indicating a good fit (F(5,49) = 34.35, *p* < 0.001).

## 4. Discussion

To my knowledge, this is the first study to examine the relationship between fatigue and self-perception of constipation among community-dwelling older adults during the COVID-19 pandemic. Fatigue is one of Fried et al.’s frailty phenotype criteria, which also include low activity, weakness, slowness, and shrinking [31]. During a pandemic, a reduction in physical activity due to lockdown and other containment measures to prevent the spread of infectious diseases, as well as muscle weakness, are common problems among older people [32]. In addition, prior research has reported that depressive symptoms and fatigue decreased the odds of maintaining sufficient physical activity during the COVID-19 pandemic [33]. This suggests that older adults are more likely to fall into a cycle of frailty during the COVID-19 pandemic. Improving self-perception of constipation, which is associated with reduced activity and food intake during a pandemic, may also help reduce fatigue and improve physical activity to break the cycle of frailty.

Fatigue is subjective and difficult to observe. Although the self-perception of constipation is also subjective, in this study, a single question was used (Do you often have constipation?). This question is easy to answer. As the first step toward focusing on one’s own health status, paying attention to whether or not one perceives constipation is a simple and useful method. If further investigation finds no underlying pathology that could be associated with both conditions, regular exercise, adequate fluid and fiber intake, and dietary modifications are the initial therapeutic approaches for primary constipation [34]; however, this is not always possible in older people and recent research has reported that specific guidelines are required for older adults [35]. In particular, there is insufficient evidence for increased exercise and fluid intake; however, there are reports that more than 30 g of fiber intake can be recommended for the symptoms of constipation [36]. According to the National Health and Nutrition Survey (2018), the average dietary fiber intake of Japanese adults is 15 g/day, with vegetables being the highest source at 5.4 g, cereals at 3.1 g, and fruits at 1.4 g [37]. The ideal target of at least 20 g/day for males > 65 years old and 17 g/day for females > 65 years old was recommended [37]. In the large bowel, large insoluble fiber particles mechanically irritate the gut mucosa, stimulating water and mucous secretion, and the high water-holding capacity of gel-forming soluble fiber resists dehydration [38]. Fiber is found in fruits, grains, vegetables, seeds, nuts, and legumes. In particular, dried prunes have both soluble and insoluble fiber in almost a 1:1 ratio and dried prunes have about 6 g/100 g fiber [39]. Recent study has found that flaxseed flour is more effective in increasing defecation frequency than lactulose [40] and psyllium [41]. Fiber improves symptoms of constipation, but studies have not yet clearly identified a particular source of fiber that works best for functional constipation [36].

Reduced physical activity is thought to be one of the causes of constipation in older adults, with reports suggesting that active individuals are less likely to experience constipation than those who have a sedentary lifestyle [42]. However, it may not have been easy to increase levels of physical activity during the COVID-19 pandemic; moreover, exercise itself may be difficult due to frailty. Therefore, effective drug interventions should also be considered when treating constipation among older adults [35]. Laxatives are the mainstay of pharmacological treatment for patients who do not respond to lifestyle or dietary modification [34]. As for laxatives, research has shown that both nonstimulant and stimulant laxatives provided increased relief for functional constipation symptoms, compared to a placebo [43]. Nevertheless, few studies have been conducted on the quality of life of people who use laxatives; therefore, further research is needed.

Moreover, research has shown that the prevalence of functional gastrointestinal disorders in children was higher during the COVID-19 pandemic [44]. In addition, patients with common functional gastrointestinal disorders and dysmotility have been reported to have more gastrointestinal symptoms during the COVID-19 epidemic [45]. The impact of activity restrictions on community dwellers during a pandemic remains largely unknown and further research is needed.

This study had four limitations. First, the generalizability of this study was limited because of its small sample size. This study was a pilot study which requires further research. Second, other potentially related factors, such as patients’ prior illnesses or medication use, were not examined in this study. Third, the proportion of those who self-perceived as constipated in this study and those who met the diagnostic criteria for FC or IBS according to Roma III was unknown, and may further include organic constipation, drug-induced constipation, and constipation secondary to an underlying disease. We also do not know how to distinguish between chronic and recent constipation. Fourth, this study analyzed valuable data from older adults who attended community gatherings that were fortunate to be held during the COVID-19 pandemic, but not before or after the pandemic, due to the status of the spread of the infection.

## 5. Conclusions

This study found a relationship between fatigue and the self-perception of constipation among community-dwelling older adults during the COVID-19 pandemic. The results showed that community-dwelling older adults with the self-perception of constipation reported higher fatigue, compared with those without the self-perception of constipation. Therefore, there is a need to develop self-management strategies that older people can use to improve the self-perception of constipation.

## Figures and Tables

**Table 1 ijerph-19-08406-t001:** Characteristics of participants.

All (*n* = 71)	Mean ± SD
Age	78.3 ± 5.5
FACIT-Fatigue (fatigue)	40.2 ± 9.7
Motor fitness scale (MFS)	10.6 ± 3.6
GDS-15 (depression)	4.0 ± 3.9
	***n* (%)**
Constipation	
Yes	40 (56.3%)
Sometimes	20 (28.2%)
No	10 (14.1%)
NA	1 (1.4%)
Living alone	
Yes	21 (29.6%)
No	50 (70.4%)
Frequency of exercise	
Usually	37 (52.1%)
Occasionally	26 (36.6%)
Hardly ever	8 (11.3%)
Never	0 (0.0%)
Economic satisfaction	
Very satisfied	17 (23.9%)
Sufficiently satisfied	28 (39.4%)
Neutral	11 (15.5%)
Somewhat satisfied	8 (11.3%)
Not sufficiently satisfied	5 (7.0%)
NA	2 (2.8%)
Subjective memory impairment	
Yes	9 (12.7%)
No	61 (85.9%)
NA	1 (1.4%)
Subjective health	
Very good	4 (5.6%)
Good	13 (18.3%)
Regular	44 (62.0%)
Poor	10 (14.1%)
Very poor	0 (0.0%)

NA = missing value.

**Table 2 ijerph-19-08406-t002:** Correlation analysis between fatigue and explanatory variables.

Variables	Mean ± SD	1	2	3	4	5	6	7	8	9	10
1. Fatigue	39.79 ± 9.43	1	−0.371 **	0.085	0.432 **	−0.750 **	0.174	0.095	0.461 **	0.020	−0.369 **
2. Age	78.29 ± 5.51		1	0.163	−0.272 *	0.322 *	0.025	0.013	−0.163	−0.054	−0.082
3. Living alone or not	0.30 ± 0.46			1	0.144	−0.049	0.200	−0.158	0.415 **	0.007	−0.043
4. MFS	10.72 ± 3.50				1	−0.208	0.157	0.023	0.389 **	−0.095	0.049
5. GDS-15	4.27 ± 4.41					1	−0.111	−0.002	−0.348 **	−0.223	0.356 **
6. Frequency of exercise	2.41 ± 0.69						1	−0.037	0.299 *	0.028	−0.092
7. SMI	0.13 ± 0.34							1	0.034	0.006	−0.072
8. Subjective health	2.15 ± 0.763								1	0.011	−0.196
9. Economic satisfaction	2.64 ± 1.19									1	−0.278 *
10. Constipation	0.57 ± 0.73										1

* *p* < 0.05, ** *p* < 0.01.

**Table 3 ijerph-19-08406-t003:** Relationship between pull factor and fatigue.

Dependent Variable	Independent Variables (β)	Intercept (95% CI)	B (95% CI)	Beta (β)	t	Sig.	Duebin Watson	Adjusted R^2^
Fatigue								
	Age	40.45 (38.28, 42.61)	−3.24 (−5.21, −1.26)	−0.37	−3.27	<0.01 **	2.237	0.125
	Living alone or not	39.62 (36.86, 42.38)	1.81 (−3.26, 6.87)	0.19	0.71	0.479	1.941	−0.007
	MFS	27.33 (20.05, 34.61)	1.21 (0.56, 1.86)	0.45	3.71	<0.001 ***	1.838	0.173
	GDS-15	47.77 (45.26, 50.27)	−1.93 (−2.38, −1.48)	−0.78	−8.63	<0.001 ***	1.507	0.555
	Frequency of exercise	34.23 (25.86, 42.61)	2.46 (−0.89, 5.80)	0.25	1.47	0.147	2.066	0.016
	SMI	39.79 (37.28, 42.30)	2.77 (−4.23, 9.77)	0.28	0.79	0.433	1.975	−0.006
	Subjective health	26.91 (20.45, 33.37)	6.15 (3.31, 8.99)	0.63	4.32	<0.001 ***	2.105	0.201
	Economic satisfaction	39.48 (33.68, 45.28)	0.17 (−1.84, 2.18)	0.02	0.17	0.867	1.919	−0.015
	Constipation	42.83 (40.07, 45.60)	−4.90 (−7.89, −1.92)	−0.37	−3.28	<0.01 **	1.866	0.124

** *p* <0.01, *** *p* <0.001.

**Table 4 ijerph-19-08406-t004:** Multiple regression analysis between independent variables and fatigue.

Stage	Independent Variables (β)	B (95% CI)	Beta (β)	t	Sig.	Duebin Watson	Adjusted R^2^	F	*p*-Value
Stage 1	Intercept (95% CI)	39.90 (31.44, 48.36)	−0.32	9.50	<0.001 ***				
	Age	−1.93 (−3.36, −0.49)	−0.22	−2.71	<0.01 **				
	Living alone or not	−0.58 (−4.19, 3.04)	−0.06	−0.32	0.750				
	MFS	0.90 (0.44, 1.36)	0.33	3.97	<0.001 ***				
	GDS-15	−1.35 (−1.78, −0.92)	−0.55	−6.31	<0.001 ***				
	Frequency of exercise	−0.03 (−2.31, 2.25)	−0.00	−0.02	0.981				
	SMI	1.97 (−1.92, 5.86)	0.20	1.02	0.313				
	Subjective health	1.34 (−1.02, 3.70)	0.14	1.15	0.257				
	Economic satisfactory	−1.68 (-2.92, −0.45)	−0.21	−2.75	<0.01 **				
	Constipation	−4.15 (−6.32, −1.97)	−0.31	−3.84	<0.001 ***	1.388	0.749	18.88	*p* < 0.001 ***
Stage 2	Intercept (95% CI)	39.85 (32.34, 47.36)	−0.32	10.68	<0.001 ***				
	Age	−1.93 (−3.34, −0.51)	−0.22	−2.74	<0.01 **				
	Living alone or not	−0.58 (−4.14, 2.99)	−0.06	−0.33	0.745				
	MFS	0.90 (0.45, 1.35)	0.33	4.01	<0.001 ***				
	GDS-15	−1.35 (−1.77, −0.92)	−0.55	−6.39	<0.001 ***				
	SMI	1.98 (−1.86, 5.81)	0.20	1.04	0.305				
	Subjective health	1.34 (−0.96, 3.63)	0.14	1.17	0.265				
	Economic satisfactory	−1.68 (−2.90, −0.47)	−0.20	−2.79	<0.01 **				
	Constipation	−4.15 (−6.29, −2.00)	−0.31	−3.89	<0.001 ***	1.390	0.754	21.71	*p* < 0.001 ***
Stage 3	Intercept (95% CI)	40.05 (32.72, 47.39)	−0.30	10.99	<0.001 ***				
	Age	−1.97 (−3.35, −0.60)	−0.22	−2.88	<0.01 **				
	MFS	0.90 (0.46, 1.35)	0.33	4.09	<0.001 ***				
	GDS-15	−1.35 (−1.77, −0.93)	−0.55	−6.46	<0.001 ***				
	SMI	2.10 (−1.61, 5.82)	0.20	1.14	0.261				
	Subjective health	1.18 (−0.87, 3.23)	0.14	1.16	0.254				
	Economic satisfactory	−1.69 (−2.90, −0.49)	−0.20	−2.84	<0.01 **				
	Constipation	−4.14 (−6.27, −2.02)	−0.31	−3.92	<0.001 ***	1.394	0.759	25.27	*p* < 0.001 ***
Stage 4	Intercept (95% CI)	40.33 (32.99, 47.67)	−0.27	11.05	<0.001 ***				
	Age	−1.97 (−3.35, −0.59)	−0.22	−2.87	<0.01 **				
	MFS	0.91 (0.46, 1.35)	0.33	4.09	<0.001 ***				
	GDS-15	−1.35 (−1.77, −0.92)	−0.55	−6.41	<0.001 ***				
	Subjective health	1.21 (−0.85, 3.27)	0.12	1.18	0.244				
	Economic satisfactory	−1.70 (−2.90, −0.49)	−0.21	−2.83	<0.01 **				
	Constipation	−4.24 (−6.36, −2.11)	−0.32	−4.01	<0.001 ***	1.353	0.757	29.09	*p* < 0.001 ***
Stage 5	Intercept (95% CI)	42.48 (36.10, 48.86)	−0.00	13.38	<0.001 ***				
	Age	−2.04 (−3.42, −0.67)	−0.23	−2.98	<0.01 **				
	MFS	0.99 (0.57, 1.41)	0.37	4.75	<0.001 ***				
	GDS-15	−1.39 (−1.81, −0.98)	−0.57	−6.76	<0.001 ***				
	Economic satisfactory	−1.75 (−2.95, −0.54)	−0.21	−2.92	<0.01 **				
	Constipation	−4.49 (−6.58, −2.40)	−0.34	−4.32	<0.001 ***	1.385	0.755	34.35	*p* < 0.001 ***

** *p* < 0.01, *** *p* < 0.001.

## Data Availability

Data are available from the author upon reasonable request and with permission from the Ethics Committee of the University of Hyogo.

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
