# Peer review of "Relationship between Fatigue and Self-Perception of Constipation in Community-Dwelling Older Adults during the COVID-19 Pandemic"

_ijerph, 2022, doi:10.3390/ijerph19148406_

Round 1

Reviewer 1 Report

While some of my observations in my original review remain I note with thanks that the author has made an effort to revise the manuscript responsively. While the submitted revision shows added material, it does not show deleted material from the original. For instance, the description 'cross sectional' has been removed, which is a little concerning, since the study remains cross-sectional, and we still find temporal ambiguity. I appreciate and acknowledge the new paragraphs that have been inserted, and I note that the tables have been disentangled. These are clearer. My concerns about the use of 3- 4- and 5- point Likert-like scales in the same tool remain, and the author does not address this issue, or the deleted material, in the cover letter. It would have been helpful to have a more comprehensive cover letter that more clearly details the author's responses to the original review comments.

All of that said, the author appears to have softened the recommendations and provided a clearer rationale for self-management, which makes the intent of the paper clearer.

There are minor presentation issues such as the alignment of columns in the tables that I will leave to the proofreaders.

Author Response

I thank you for your feedback and healpful suggestions. I have revised the paper significantly in response to the reviewers' comments and I hope the revised version in acceptable for publication. I have highlited the revised text in yellow in the main manuscript. 

I added the sentence "cross-sectional survey was conducted among" (line 12-13) and "We conducted a cross-sectional survey." (line 70)

As for the use of 3-, 4-, 5-point Likert scales in this study, I know that there is a lot of controversy, but all of these items have been used in other surveys before, so I do not believe that they will have a significant impact on the results. I thank you for your very significant remarks.

Reviewer 2 Report

1) General comments

Dr. Hayashi revised ‘Relationship between fatigue and self-perception of constipation in community-dwelling older adults during the COVID-19 pandemic’.

Thank you for your reply. I read your responses for my questions.

Your answers are precisely good, and I understood your points of view in your study.

I appraise your investigation. I think that it is suitable for this journal.

Author Response

 I thank you for your feedback and helpful suggestions. 

Reviewer 3 Report

Hayashi et al has done a study aimed to examine the relationship between fatigue and self-perception of constipation in community-dwelling older adults during the COVID-19 pandemic. The study has showed that the fatigue score in community-dwelling older adults who had self-perception of constipation was significantly lower (i.e., they had higher fatigue). Although this study has a few limitations, such as its small sample size, single-center, it can be one of the other studies analyzing the relationship between fatigue and self-perception of constipation.

Author Response

 I thank you for your feedback and helpful suggestions. 

This manuscript is a resubmission of an earlier submission. The following is a list of the peer review reports and author responses from that submission.

Round 1

Reviewer 2 Report

1) General comments

Dr. Hayashi investigated ‘Relationship between fatigue and functional constipation in community-dwelling older adults during the COVID-19 pandemic’. The reviewer has some comments.

  1. In Introduction, the author describes the aim of this study, “during the COVID-19 pandemic”. In Results and Discussion, the author should explain how different your results between during the COVID-19 pandemic and non-pandemic period.
  2. In Material and Methods, the author should explain characteristics of participants more detail such as gender, comorbidity, concomitant use of medicine, and so on.
  3. Please explain how constipation and fatigue are related in Introduction and Discussion.

Reviewer 3 Report

Thank you for the opportunity review ‘Relationship between fatigue and functional constipation in community-dwelling older adults during the COVID-19 pandemic’. The article reports an analysis of a survey of 97 older people living in the community isolating during the Covid 19 pandemic, and attempts to consider possible predictors and confounders of fatigue. The author used a stepwise regression analysis to analyse develop a predictive model, and concludes that functional constipation is most closely associated with fatigue; the discussion and conclusions propose treating the constipation in order to address the fatigue and possibly associated frailty.

This paper puts me in mind of one of my first epidemiology courses where one of the first things we learned was ipso hoc non ergo propter hoc (apologies for the Latin, but that’s how it was presented): that is, just because two things appear together does not mean that they are related to one another. There can be no temporal relationship in a cross-sectional study (which this study was), so we do not know whether the fatigue preceded the functional constipation, or the constipation preceded the fatigue: the literature suggests that fatigue is a common sequela of constipation. Secondly, because of the nature of the survey the author cannot know whether both constipation and fatigue are related to a ‘hidden’ third factor—say bowel cancer, which could presumably result in both symptoms. The author notes only two limitations to the study (sample size, and that other illness or medication use were not examined). The sample size could be addressed by claiming this was a pilot study which required further study. The second limitation is a considerable one. Based on this study, if a health provider were to address fatigue only by treating functional constipation, I think there would be considerable risk of overlooking more significant causes; perhaps the author would even agree with me on this. But if the author does not want the paper to be used in this way, then why do they seek to publish this finding? For me this presents fundamental challenge to the publication of this paper.

There are a number of smaller issues throughout the paper that require attention: in Line 43, ‘comorbidity, self-related health’ are not meaningful terms in the list presented.  The statistical tests that were carried out will all have assumptions associated with them. One of the most common assumptions is normality of distribution. The author does not say whether the sample responses were normally distributed, or if not, what was done to compensate for that. I found it remarkable that three-point, four-point and five-point Likert-types scales were used in the same survey, and I have to wonder how participants coped with this range of response types. Line 75 says that ‘statistical significance was set a P<.05’—I have seen this more commonly phrased as ‘a threshold alpha was set at α=.05’. More importantly, in Line 91 ‘the participants who answered the above question [on functional constipation] with a ‘yes’ were considered to be ‘constipated’, but in lines 126-27 ‘a total of 30 people who answered “yes” and “sometimes” were included in the constipation group’. The different criterion for inclusion in the constipation group is not explained. Different numbers will change the analysis considerably.

I found myself very confused between Tables 2 and 3. Line 149 says that the results for confounders is found in Table 2, but I don’t find that in Table 2. The rest of that paragraph (lines 150-153) refer to results that are not found in the tables; the relationship between the tables and the text is unclear to the reader. Tables should summarise the text, not present new or different information. The text should be able to stand alone. Is the difference in the variance the result of the differences in the number of participants included in the constipation group?

Finally, as noted above, I am concerned that in the conclusion the author seems to propose that self-management strategies are all that is required when presenting with functional constipation and fatigue. (lines 213-216), without proposing further investigation. Even adding the words ‘if further investigation finds no underlying pathology that could be associated with both conditions’ would go some way to mitigate recommendation. Without them, or at least without further elaboration, the recommendation seems to me to present considerable risk.

The formatting between text and tables was somewhat difficult to follow, but I will leave this issue to the editorial staff. The references are reasonable and relevant.